# The Elephant in the Room: A Cross-Sectional Study on the Stressful Psychological Effects of the COVID-19 Pandemic in Mental Healthcare Workers

**DOI:** 10.3390/brainsci12030408

**Published:** 2022-03-19

**Authors:** Alessandra Minelli, Rosana Carvalho Silva, Stefano Barlati, Marika Vezzoli, Sara Carletto, Cinzia Isabello, Marco Bortolomasi, Gabriele Nibbio, Jacopo Lisoni, Valentina Menesello, Giulia Perusi, Vivian Accardo, Giacomo Deste, Antonio Vita

**Affiliations:** 1Department of Molecular and Translational Medicine, University of Brescia, 25121 Brescia, Italy; alessandra.minelli@unibs.it (A.M.); r.carvalhosilva@unibs.it (R.C.S.); marika.vezzoli@unibs.it (M.V.); v.accardo@unibs.it (V.A.); 2Genetics Unit, IRCCS Istituto Centro San Giovanni di Dio Fatebenefratelli, 25125 Brescia, Italy; 3Department of Mental Health and Addiction Services, ASST Spedali Civili of Brescia, 25123 Brescia, Italy; jacopo.lisoni@gmail.com (J.L.); valentina.menesello@gmail.com (V.M.); giuliaperusi@gmail.com (G.P.); giacomodeste@mac.com (G.D.); antonio.vita@unibs.it (A.V.); 4Department of Clinical and Experimental Sciences, University of Brescia, 25121 Brescia, Italy; gabriele.nibbio@gmail.com; 5Department of Neuroscience “Rita Levi Montalcini”, University of Torino, 10124 Torino, Italy; sara.carletto@unito.it; 6Mental Health Service of Susa, ASL TO3, Susa, 10093 Torino, Italy; cisabello@aslto3.piemonte.it; 7Psychiatric Hospital “Villa Santa Chiara”, 37142 Verona, Italy; marcobortolomasi.vr@gmail.com

**Keywords:** COVID-19, mental healthcare workers, stressful life events, anxiety, depression, stress-related disorder

## Abstract

Despite extensive research on COVID-19’s impact on healthcare workers, few studies have targeted mental health workers (MHWs) and none have investigated previous traumatic events. We investigated psychological distress in MHWs after the first lockdown in Italy to understand which COVID-19, sociodemographic, and professional variables represented greater effects, and the role of previous trauma. The survey included sociodemographic and professional questions, COVID-19 variables, and the questionnaires Life Events Checklist for DSM-5 (LEC-5), Impact of Event Scale—Revised (IES-R), and Depression Anxiety Stress Scales 21 (DASS-21). On the 271 MHWs who completed the survey (73.1% female; mean age 45.37), we obtained significant effects for contagion fear, experience of patients’ death, increased workload, and worse team relationship during the first wave. Nurses were more affected and showed more post-traumatic stress symptoms, assessed by IES-R, and more depressive, anxiety, and stress symptoms, assessed by DASS-21. The strongest risk factors for distress were greater age, professional role, increased workload, worse team relationship, and separation from family members. Previous experience of severe human suffering and unwanted sexual experiences negatively impacted IES-R and DASS-21 scores. Being a psychiatrist or psychologist/psychotherapist and good team relationships were protective factors. Recent but also previous severe stressful events might represent relevant risk factors for distress, reducing resilience skills. Identifying vulnerable factors and professional categories may help in the development of dedicated measures to prevent emotional burden and support psychological health. Highlights: Psychological distress in mental health workers in the COVID-19 pandemic is more frequent in nurses, who experience more depression, anxiety, and post-traumatic stress symptoms. Previous and recent stressful events are risk factors for distress and should guide intervention strategies.

## 1. Introduction

The COVID-19 pandemic is impacting everyone’s life, particularly those of healthcare workers (HCWs). Healthcare professionals are experiencing a strong discomfort related to their direct involvement in the resolution of this public health emergency and in ethical challenges in often unadjusted work contexts [1,2]. During the first and second waves, as it occurred in the North of Italy from the end of February to the end of March 2020 and from the end of October to the end of November 2020, respectively, HCWs faced a great number of deaths among patients and colleagues, as well as their own exposure to this potentially fatal infection, with consequent mental and physical exhaustion, burnout, depression, anxiety, grief, and insomnia [3,4,5,6]. Deterioration in workers’ mental health results from increased work-related stress caused by these factors, but also from exposure to specific traumatic experiences [7]. Literature data indicate that COVID-19-involved HCWs suffer from different kinds of psychological distress. For instance, a Chinese study in Wuhan, the epicenter of the pandemic, revealed prevalence rates of 28.6% for moderate and severe mental health disturbances and 34.4% for mild psychological distress among HCWs [8]. A Chinese study in Shanghai during the first outbreak showed a prevalence of emotional distress of 48%, with women and nurses being more affected [9]. Another Chinese study reported that 12.5% of professionals suffered from anxiety, with higher scores for those who had direct contact treating infected patients [10]. Also, employment relationships were highly impacted by the pandemic, and some evidence showed the importance of maintaining a healthy work environment, avoiding role conflicts and burnout syndromes during the pandemic context [11]. The World Health Organization (WHO) underlines that COVID-19 represents a long-term emergency—“This is not a sprint, it’s a marathon”—and epidemiologic predictions indicate that this pandemic will double psychiatric morbidity among HCWs. Recent meta-analyses show high percentages of prevalence ranging from 15 to 30% for anxiety disorders, from 15 to 31% for depression, and from 24 to 44% for insomnia and sleep disturbances, with a current incidence of about 21% for PTSD [4,5,6].

In this context, mental health workers (MHWs) may experience particularly severe consequences related to the pandemic, since they are already involved in complex situations, dealing with frail and difficult-to-manage patients, hence being more vulnerable to work-related stress. MHWs face the increased mental health needs of both the general population, stressed out by current limitations, and patients with pre-existing mental disorders experiencing worsening symptoms [1,12]. Restrictive measures introduced in Italy by health authorities in March 2020 have deeply affected the work practices of mental health services: many services have been forced to quickly adopt new ways of delivering mental health services (e.g., teletherapy, consultation via phone or mail) without proper training. Inpatient units have been organized with areas for COVID-19-positive patients with acute mental disorders, whereas home care and remote psychosocial interventions are being provided for urgent cases [13,14,15,16,17]. Despite the large amount of research addressing various aspects of pandemic impacts on HCWs, to date, only two reports have specifically targeted MHWs. An Italian study showed that approximately 31% of the participants had a severe score in at least one burnout dimension, 12% showed moderate or severe levels of anxiety, and 7% had moderate or severe levels of depression, with different patterns of distress for inpatient versus outpatient service MHWs. This suggests that, although the impact of the pandemic on MHWs’ distress is mild, a significant number of workers experience severe levels of depersonalization and anxiety [18]. A Chinese study involving community MHWs reported a higher risk of depression and anxiety among those who provided service for suspected COVID-19 cases who were quarantined; on the contrary, it was shown that professionals receiving psychiatric training had higher positive emotion and self-efficacy [19]. Both general and job-specific stressors are at play, and workplace protective factors and increases in coping skills linked to the professional role may mitigate the negative impacts of pandemic-related stress. 

As demonstrated in the literature, previous traumatic events represent risk factors for psychological distress, particularly PTSD symptoms, with a cumulative effect that can exacerbate the negative effects of previous stressors [20,21]. Nonetheless, to date, no study has investigated the effect of previous stressful life experiences on psychological distress due to COVID-19 emergency in HCWs. 

To fill this gap, this study aimed at: (1) investigating which COVID-19 variables, including positive test status, having an infected family member, fear of contagion, death of a loved one or death a patient by the infection, alterations in workload, team relationship changes, and family divided due to the current pandemic, had a psychological effect in a group of MHWs in the North of Italy after the first lockdown imposed by the pandemic (i.e., March to May 2020); (2) addressing which sociodemographic and professional variables had a psychological effect in the population investigated; and (3) understanding whether previous stressful experiences may have a negative influence on the emotional distress experienced during the pandemic in the studied cohort.

## 2. Materials and Methods

### 2.1. Participants

The inclusion criteria were being a MHW employed at public and/or private facilities (e.g., hospital wards, residential facilities, and community mental health centers) in one of the three northern Italian regions most impacted by the first COVID-19 outbreak (i.e., Piedmont, Lombardy, and Veneto) and aged ≥18 years. 

Data collection occurred from 28 June to 10 August 2020 (see Appendix A), using a Google Form online survey. An invitation e-mail was initially sent to potential participants, and further subjects were recruited using a snowball sampling procedure by asking participants to disseminate the survey link among their colleagues. A detailed file containing the invitation e-mail and the survey details is available in the Appendix A in their original Italian versions and in the English language. 

The study was conducted in accordance with the Declaration of Helsinki and was approved by the Ethics Committee of the ASST Spedali Civili of Brescia, Italy (protocol NP4172). Written electronic informed consent was obtained by asking all participants to click a button at the beginning of the online survey informing of their consent to participate.

### 2.2. Assessment

The online survey took about 15 min to complete and contained two sections. The first included questions about sociodemographic factors (age, gender, educational level, marital and living status), professional information (profession, workplace, mental healthcare facility, years of professional experience), and data about COVID-19 exposure. 

In the second section, validated self-report questionnaires were administered:-Life Events Checklist for DSM-5 (LEC-5) [22] is a 17-item self-report questionnaire to assess exposure to traumatic events. Respondents are asked to indicate whether they have experienced, witnessed, or learned about 17 traumatic events. As the LEC-5 collects information about potentially traumatic experiences a person has experienced, there is no total or composite score for this scale.-Impact of Event Scale—Revised (IES-R) [23] is a 22-item questionnaire to evaluate subjective distress caused by traumatic events. The instrument comprises three subscales assessing intrusion (8 items), avoidance (8 items), and hyperarousal (6 items) symptoms. For this study, participants were asked to refer to the COVID-19 emergency when answering the questionnaire. IES-R items are rated on a 5-point scale ranging from 0 (“not at all”) to 4 (“extremely”), with a total score ranging from 0 to 88. In line with previous research, a score of ≥33 likely indicates the presence of PTSD [24].-Depression Anxiety Stress Scales 21 (DASS-21) [25] is a 21-item self-report questionnaire designed to measure states of depression, anxiety, and stress, with 7 items for each subscale. Participants are asked to score each item on a scale from 0 (“did not apply to me at all”) to 3 (“applied to me very much”). Sum scores are computed by adding up the scores on the items per (sub)scale and multiplying them by a factor of 2. Sum scores for each subscale may range between 0 and 42, with higher scores indicating higher depression/anxiety/stress levels.

### 2.3. Outcomes

The primary outcome is the IES-R total score in relation to the other variables.

Secondary outcomes include IES-R subscales (intrusion, avoidance, and hyperarousal) in relation to the other variables. 

Tertiary outcomes include DASS dimension scores, as well as the effects of previous stressful experiences measured by LEC-5, in relation to the other variables. 

### 2.4. Statistical Analysis

In the descriptive statistics, quantitative variables are shown as the means and standard deviations (SD), median (IQR), and range (min–max), while qualitative variables are shown as counts and percentages. Comparisons among subgroups of subjects, obtained by stratifying in respect to a factor variable, were made using the Kruskal test for medians and Fisher’s exact test for proportions. Pairwise comparisons were adjusted by the Hommel method.

Spearman correlations between a couple of quantitative variables are visualized by means of a correlation plot, where blue and red circles correspond to positive or negative correlation, respectively. The circle diameter and color intensity are proportional to the magnitude of the Spearman coefficient, and if there are black crosses on them, correlations are not significantly different from zero (*p*-values of >0.05). The correlation matrix was reordered according to the hierarchical cluster analysis on the quantitative variables. 

To understand which variables have a strong impact on the IES-R total, as well as the DASS scale and subscale scores, random forest (RF), a machine learning (ML) approach that grows many regression trees, was used [26]. This non-parametric method belonging to the ensemble models is able to deal with variables of different natures (qualitative and quantitative) and models nonlinear relationships between the outcome and the covariates. Missing values were imputed with the MissForest algorithm, a non-parametric method dealing with mixed-type variables. This imputation method considers complex interactions and nonlinear relationships and is robust to noisy data and multicollinearity.

In detail, the outcomes (IES-R total, DASS scale and subscale scores) were modelled by an RF with the following covariates of different nature (quantitative and qualitative): age, professional roles, variables related to COVID-19 exposure, and the 17 LEC items [27]. This algorithm provides a measure of importance of each feature called the relative Variable Importance Measure (relVIM), ranging from the most (relVIM = 100) to the least important variable [28]. The RF is visualized by means of a bar plot and could be used for variable selection considering only those variables with a relVIM value of >20. 

Multivariable logistic regression analyses were used to detect risk factors in association with symptom severity. Participants scoring greater than the cutoff threshold were characterized as having at least moderate symptoms; consequently, the IES-R total and DASS scale and subscale scores were dichotomized (IES-R total ≥ 26; DASS—depression subscale ≥ 14; DASS—anxiety subscale ≥ 10; DASS—stress subscale ≥ 19) based on literature values [25]. Four multivariate logistic regression models were estimated to detect which variables (age, professional role, variables related to COVID-19 exposure, and the 17 LEC items) are associated with higher levels of the scales inspected (participants scoring greater than the cutoff were characterized as having at least moderate symptoms). The results report the odds ratio (OR), corresponding 95% confidence interval (CI) and *p*-values only in correspondence to significant covariates in the model.

## 3. Results

### 3.1. Demographic, Professional, and Stressful Life Event Characteristics

Two hundred and seventy-one MHWs completed the survey. The sociodemographic and professional descriptions are reported in Table 1, whereas the prevalence of stressful events, as assessed by LEC-5, is shown in Appendix A. Approximately two-thirds of the participants work in outpatient services and most of them are female (73.1%), with an average age of 45.4 years (SD = 10.9) and average period of education of 17.7 years (SD = 3.9). All mental health professional categories were represented, and most of the participants have at least 10 years of professional experience. The participants declared to have experienced several traumatic events as assessed by LEC-5. Indeed, several items had a prevalence higher than 20%, including natural disaster, transportation accident, physical assault, illness injury, severe human suffering, or any stressful life-time events (for prevalence details, see Appendix A).

### 3.2. Scores of Measurements and Associated Factors

Analyses of COVID-19 variables and professional roles revealed significant differences in relation to work characteristics (Table 2). Significant effects were obtained for contagion fear, experience of patients’ death, increased workload, and worse team relationships. For all these variables, nurses were more affected in comparison to other MHWs. 

Concerning symptom assessments, we found significant positive correlations (*p*-values of <0.01) between the IES-R and DASS dimensions (see the correlation plot in Appendix A), demonstrating a high internal coherence and good quality congruence in response to the survey.

Significant associations were found between IES-R and DASS dimension scores with professional roles, showing significant differences in relation to the work characteristics (Table 3 and Table 4). Pairwise comparisons showed that nurses experienced more stress, anxiety, and depressive symptoms compared to other MHWs.

### 3.3. Risk Factors of Measured Outcomes

Finally, the relVIM identified the variables with stronger impacts on the predictions of the IES-R total score and for the three DASS subscales. 

The results of the RF (Table 5) showed that the most significant influences for all outcomes were increased age, kind of professional role, workload and team relationship changes, and separation from family members. Stressful life events assessed by LEC-5 had a strong impact on IES-R total scores and on DASS subscales, including experiences of severe human suffering (LEC-5 Item 13) and unwanted sexual experiences (LEC-5 Item 9). For detailed lists of variables of importance for each outcome, see Appendix A.

Multivariable logistic regression analysis showed that stressful events were associated with moderate or severe symptoms of depression, anxiety, and distress. In particular, risk factors associated with moderate or severe symptoms indicated by IES-R total score were age (*p* = 0.0037) and increased workload (*p* = 0.0128). Having lived severe human suffering (LEC-5 Item 13) and unwanted sexual experiences (LEC-5 Item 9) (*p* = 0.0022 and *p* = 0.028, respectively) represented further risk factors. On the contrary, being a psychiatrist (or a psychiatrist trainer) and being a psychologist–psychotherapist were protective factors to moderate or severe symptoms as assessed by the IES-R total (*p* = 0.0005 and *p* = 0.0050, respectively). Having a great team relationship during the emergency reduced symptom severity (*p* = 0.034).

Risk factors associated with moderate or severe symptoms on the DASS Depression subscale were exposure to toxic substances (LEC-5 Item 5; *p* = 0.0016), unwanted sexual experiences (LEC-5 Item 9; *p* = 0.0273), and severe human suffering (LEC 5 Item 13; *p* = 0.0079), whereas being a psychologist or psychotherapist represented a protective factor (*p* = 0.0254).

A risk factor associated with moderate or severe symptoms on the DASS Anxiety subscale was having lived unwanted sexual experience (LEC-5 Item 9; *p* = 0.0082), whereas being a psychiatrist (or a psychiatrist trainer) or a psychologist/psychotherapist was a protective factor (*p* = 0.0211 and *p* = 0.0463, respectively).

Finally, risk factors associated with moderate or severe symptoms on the DASS Stress subscale score were having lived unwanted sexual experience (LEC-5 Item 9) and experiencing severe human suffering (LEC-5 Item 13) (*p* = 0.0051 and *p* = 0.0142, respectively). Instead, being a psychiatrist and having a great team relationship during the pandemic led to reduced stress symptom severity (*p* = 0.0388 and *p* = 0.0168, respectively).

Detailed *p*-values and ORs of the significant predictive factors for the measured outcomes are shown in Table 6.

## 4. Discussion

We evaluated which factors represent stronger psychological effects on emotional distress experienced during the COVID-19 pandemic within MHWs in a cohort composed of a wide range of professionals.

Studies assessing psychological impacts of the pandemic in HCWs are relevant since this population is especially vulnerable to its consequences. After patients themselves, the highest burden of psychological morbidity was found within HCWs in relation to the general population [29], a finding also observed in previous surveys during epidemic contexts [30,31]. Many systematic reviews were conducted in order to unravel the psychological impacts of the current pandemic on HCWs [4,32,33,34,35,36,37], some of them focusing on PTSD [38] or anxiety [36]. 

Although the literature regarding the impact of COVID-19 pandemic in general HCWs is vast, a focus on MHWs is scarce; this is one of the strengths of our work, which was dedicated to a specific cohort. One study showed a mild impact of anxiety and depression in MHWs, but with severe levels of burnout in 31% of participants and a significant number of workers reporting severe levels of depersonalization and anxiety [18]. A higher risk for depression and anxiety was found in community MHWs providing services for people with suspected infection who were in quarantine [19]. Adding to the scarce available evidence, our study has the particularity of addressing a subgroup of HCWs especially vulnerable to mental impacts imposed by the pandemic.

We characterized the population for previous stressful experiences, a factor that can increase psychological burden and decrease resilience. The participants declared having experienced several traumatic events, and some items showed a prevalence higher than 20%, including natural disaster, transportation accident, physical assault, illness injury, and severe human suffering. Stressful events represented a strong impact on distress caused by the pandemic. Previous stressful events increase the susceptibility for psychopathology, leading to greater symptom severity, longer disease course, and unfavorable outcomes [39,40,41,42,43]. In our survey, exposures to toxic substances (LEC-5 Item 5), unwanted sexual experiences (LEC-5 Item 9), and severe human suffering (LEC-5 Item 13) were risk factors of moderate or severe depressive symptoms, as assessed by the DASS Depression subscale. Unwanted sexual experience (LEC-5 Item 9) was a risk factor of moderate or severe symptoms on the DASS Anxiety subscale. Unwanted sexual experience (LEC-5 Item 9) and severe human suffering (LEC-5 Item 13) were risk factors of moderate and severe distress symptoms, as assessed by both the IES-R total score and the DASS stress subscale. These last two exposures are relevant since they are risk factors for both depressive and stress symptoms assessed by the DASS subscales, as well as for distress symptoms assessed by the IES-R total score. Our results are in line with other findings. A study aiming to evaluate distress in Italy during the initial COVID-19 pandemic showed that a history of stressful situations was associated with increased anxiety and depression in the general population, highlighting the importance of previous stressors [44]. It is known that individuals exposed to trauma suffer long-term consequences, presenting symptoms that may emerge during future stressful situations [45]. It is also known that HCWs presenting less efficient stress coping abilities are more likely to develop psychopathological sequelae, and professionals showing greater resilience levels are less affected by stressful environments [46,47]. 

Another relevant finding of our study is that at least 20% of the sample presented a possible diagnosis of PTSD. This is in accordance with other results showing that approximately 30% of MHWs presented severe scores in at least one burnout dimension, including emotional exhaustion, depersonalization, and reduced personal accomplishment [18]. Recent meta-analyses found between 11 and 73.4% of HCWs reporting post-traumatic stress symptoms during this pandemic and similar epidemics [46,48]. Other authors reported levels of psychological distress/acute stress disorders varying from 34% to 56.5% in the current pandemic [5,29,49,50], with post-traumatic stress symptoms varying from 20% to 49% [3,5,29,49]. These differences in prevalence are probably due to the use of diverse assessments to evaluate psychological impacts; the specific epidemiological contexts in which the studies were made; and the particularities of each healthcare service, including structural facilities and psychosocial support for professionals and their families, among others.

Stratifying by profession, our study found that nurses and educators/psychiatric rehabilitation therapists were the most affected, but post-traumatic symptoms were more present among nurses. Nurses were also more affected by depression, anxiety, and stress symptoms. Nearly 70% of nurses reported a notable increase in workload, and contagion fear, experience of patients’ death, and worse team relationship were more prevalent among nurses. Indeed, several meta-analytic studies have reinforced the vulnerability of certain professions, especially nurses, given their closer and more prolonged contact with infected patients and higher work pressures [35,49,51]. Previous research reported greater psychological impact suffered by nurses, due to their increased risk perception and higher maladaptive responses in consequence of greater contact with infected patients, more prolonged exposure to patients’ clinical complications and negative outcomes, and a shortage of protective equipment and facilities [46,51]. Nurses demonstrated poorer mental health outcomes compared with other HCWs during the pandemic, showing more depression, anxiety, and post-traumatic stress symptoms [51]. Moreover, nurses represented the largest affected population with higher prevalence of anxiety, depression, distress, and post-traumatic stress symptoms, especially among front-line workers [49,52,53,54,55,56,57]. Nurses and other HCWs working in mental health services are frequently exposed to stressful factors, particularly those related to the daily management of psychiatric patients, who may be difficult to manage and more demanding and usually show less compliance with infection preventive measures, including wearing masks and maintaining social distancing. Moreover, psychiatric patients are usually more vulnerable to psychological distress and to anxiety symptoms, which further complicates their management by MHWs directly involved in their care.

The major risk factors for distress and depression, anxiety, and stress symptoms were the following: greater age, nurse as a professional role, increased workload, worse working environment, separation from one’s family during the pandemic, severe human suffering (LEC-5 Item 13), and unwanted sexual experiences (LEC-5 Item 9). On the contrary, being a psychiatrist or psychologist/psychotherapist was a protective factor, decreasing the risk for depression, anxiety, and stress, showing the importance of education to develop stress coping skills for all MHWs. Indeed, in spite of being exposed to severe stress loads, overall levels of anxiety and depressive symptoms within MHWs were relatively low, especially within some professional categories, like psychiatrists and among psychologists/psychotherapists. We can speculate about the importance of developing resilience skills and coping strategies to face difficult professional situations and unforeseen circumstances in all professional categories—abilities that seemed to be already present among those having probably faced an educational environment specifically focused in and sensitive to mental health issues. 

Some limitations must be addressed in our study. For the population selection we sent electronic invitations to the population of interest and asked the participants to disseminate the survey link among their colleagues, in a snowball sampling procedure. This selection strategy presents some limitations that need to be considered. First, it does not guarantee that the sample is well representative of a larger population. Secondly, this type of selection may recruit subjects sharing the same traits and characteristics, which may homogenize the studied population. On the other hand, this selection procedure is easy to perform and enables the recruitment of a larger population of interest, suitable for our exploratory analysis. The assessments were made using self-report questionnaires, which may not completely and truly represent the real psychological impacts of a given situation, especially on sensitive questions. Moreover, no confirmation tool for DSM-5 diagnoses with clinical interviews was used to confirm the presence of an actual disorder. This limitation, presented by our study and by others, is understandable since these evaluations are not easily feasible in a pandemic situation. Also, we are aware that a myriad of other factors may have influenced the perceived psychological distress impacts, including age and time of exposure to previous traumatic events, treatment or absence of treatment for previous stressful event exposure, and other possible diagnoses of depressive and anxiety symptoms already taking place before the occurrence of the pandemic adding to current distress symptoms, to mention some of them. Future studies taking into account these considerations and variables would be valuable in order to broaden the analyses of potential influencing factors. The cross-sectional design of the study is another limitation, especially if we consider the constantly changing contexts of the pandemic situation, leading to diverse impacts on the psychological and mental health of the most involved population. Therefore, longitudinal analyses taking these variabilities into account are warranted since they are much more representative of the actual and dynamic impacts on the pandemic. Another point to consider is that the current study was addressed to a specific population in a determined region and healthcare context. Therefore, the present results cannot be extended to other contexts, different healthcare systems, or populations differentially impacted by the pandemic. 

A limitation presented by many studies directed to the evaluation of HCWs’ psychological impacts during the current pandemic is clinical assessment heterogeneity. In order to avoid this bias, we used validated questionnaires, with well-established cutoff values. Indeed, a strength of our study was the utilization of validated measurements in a well-characterized cohort—essential factors for the consistency and comparability of the results. Also, our recruitment process provided a comprehensive representation of all MHW categories present in our mental healthcare context.

## 5. Conclusions and Perspectives

Our study highlights risk factors for psychological impacts of the pandemic on MHWs, such as professional role (particularly in nursing), increased workload, worsening of work relations, age, and separation from one’s own family during the first wave, suggesting that MHWs may be a vulnerable population put under a high mental health burden during the pandemic. Identifying vulnerable age groups and professional categories may help in the development of measures to prevent emotional burden and support psychological health, educational programs targeted to the more vulnerable categories, and interventions to enhance resilience and coping strategies. 

Recommended strategies may include adequate work-related information, adjustments in work hours, healthy lifestyle, protective supplies and workplace interventions, professional counselling, and psychological support services [58]. Psychological support based on the particular needs of different staff members, anxiety control measures, and adequate support to deal with end-of-life and palliative care are needed [59,60]. This support can also include e-learning strategies, online facilities with interventions for the management of the pandemic, and a supportive environment considering the specificities of more vulnerable populations [46,59,61]. 

Moreover, it is important to consider the role of previous severe stressful events as risk factors to post-traumatic stress symptoms. For these at-risk individuals, alternative strategies could be considered, like trauma-focused psychotherapies, including trauma-focused cognitive behavioral therapy (TF-CBT) and eye movement desensitization and reprocessing (EMDR), the effective benefits of which have been shown for depressed individuals presenting exposure to early life stressful events [62,63]. 

Overall, our results may help in unraveling the less acknowledged risk factors for post-traumatic stress, including exposure to severe previous stressful events, which might decrease resilience levels and stress coping abilities. For this reason, the title refers to the “elephant in the room”, using this metaphorical expression to denote an obvious issue that needs to be addressed but is currently not being given enough attention. 

## Figures and Tables

**Table 1 brainsci-12-00408-t001:** Descriptive statistics on demographical and professional variables.

Variable	Overall (N = 271)
**Gender**	
Female	198 (73.1%)
Male	73 (26.9%)
**Age**	
Mean (SD)	45.37 (10.94)
Median (Q1, Q3)	47.00 (36.00, 55.00)
Range	22.00–67.00
**Years of education**	
Mean (SD)	17.72 (3.85)
Median (Q1, Q3)	16.00 (16.00, 23.00)
Range	8.00–23.00
**Marital status**	
Married	187 (69.1%)
Unmarried	51 (18.8%)
Divorced	31 (11.4%)
Widowed	2 (0.7%)
**Living condition**	
Cohabiting couples with sons/daughters	120 (44.3%)
Cohabiting couples	68 (25.1%)
Alone	38 (14.0%)
Sons/Daughters	18 (6.6%)
Parents	17 (6.3%)
Other conditions	10 (3.7%)
**Professional role**	
Nurse	86 (31.7%)
Psychiatrist/Training psychiatrist	60 (22.1%)
PRT/Educator	59 (21.8%)
Psychologist/Psychotherapist	44 (16.2%)
Other mental health professions	22 (8.1%)
**Workplace**	
Hospital	159 (58.7%)
Local mental health services	80 (29.5%)
Psychiatric residence	25 (9.2%)
Semi-residential centers	7 (2.6%)
**Years of professional experience**	
At most 10	74 (27.3%)
Greater than 10	197 (72.7%)

PRT: psychiatric rehabilitation therapist.

**Table 2 brainsci-12-00408-t002:** Descriptive statistics of events that happened during the analyzed COVID-19 period.

Variable	Total (N = 271)	Nurses (N = 86)	Psychiatrists/Training Psychiatrists (N = 60)	PRTs/Educators (N = 59)	Psychologists/Psychotherapists (N = 44)	Other Mental Health Professionals (N = 22)	*p*-Value
**COVID-19 positive**							
No	255 (94.1%)	77 (89.5%)	57 (95.0%)	57 (96.6%)	43 (97.7%)	21 (95.5%)	0.353 ^a^
Yes	16 (5.9%)	9 (10.5%)	3 (5.0%)	2 (3.4%)	1 (2.3%)	1 (4.5%)	
**Family member COVID-19 positive**							
No	248 (91.5%)	78 (90.7%)	55 (91.7%)	52 (88.1%)	42 (95.5%)	21 (95.5%)	0.754 ^a^
Yes	23 (8.5%)	8 (9.3%)	5 (8.3%)	7 (11.9%)	2 (4.5%)	1 (4.5%)	
**Fear of contagion**							
No	165 (60.9%)	44 (51.2%)	27 (45.0%)	46 (78.0%)	31 (70.5%)	17 (77.3%)	**<0.001**^a^*
Yes	106 (39.1%)	42 (48.8%)	33 (55.0%)	13 (22.0%)	13 (29.5%)	5 (22.7%)	
**Death of a loved one**							
No	241 (88.9%)	77 (89.5%)	48 (80.0%)	55 (93.2%)	42 (95.5%)	19 (86.4%)	0.092 ^a^
Yes	30 (11.1%)	9 (10.5%)	12 (20.0%)	4 (6.8%)	2 (4.5%)	3 (13.6%)	
**Death of a patient**							
No	240 (88.6%)	72 (83.7%)	49 (81.7%)	58 (98.3%)	43 (97.7%)	18 (81.8%)	**0.002**^a^*
Yes	31 (11.4%)	14 (16.3%)	11 (18.3%)	1 (1.7%)	1 (2.3%)	4 (18.2%)	
**Workload**							
Decreased	78 (28.8%)	11 (12.8%)	18 (30.0%)	20 (33.9%)	24 (54.5%)	5 (22.7%)	**<0.001**^a^*
Unchanged	77 (28.4%)	16 (18.6%)	24 (40.0%)	18 (30.5%)	10 (22.7%)	9 (40.9%)	
Increased	116 (42.8%)	59 (68.6%)	18 (30.0%)	21 (35.6%)	10 (22.7%)	8 (36.4%)	
**Team relationship**							
Got worse	98 (36.2%)	34 (39.5%)	21 (35.0%)	27 (45.8%)	13 (29.5%)	3 (13.6%)	**0.043**^a^*
Unchanged	112 (41.3%)	34 (39.5%)	23 (38.3%)	23 (39.0%)	16 (36.4%)	16 (72.7%)	
Improved	61 (22.5%)	18 (20.9%)	16 (26.7%)	9 (15.3%)	15 (34.1%)	3 (13.6%)	
**Family divided**							
No	186 (68.6%)	60 (69.8%)	37 (61.7%)	38 (64.4%)	34 (77.3%)	17 (77.3%)	0.529
Yes	38 (14.0%)	12 (14.0%)	11 (18.3%)	6 (10.2%)	7 (15.9%)	2 (9.1%)	
I live alone	35 (12.9%)	12 (14.0%)	8 (13.3%)	10 (16.9%)	3 (6.8%)	2 (9.1%)	
Unknown	12 (4.4%)	2 (2.3%)	4 (6.7%)	5 (8.5%)	0 (0.0%)	1 (4.5%)	

^a^ Fisher exact test. * Corrected (Hommel method) pairwise comparison result: “Fear of contagion”—Nurses and Psychiatrists/Training psychiatrists vs PRTs/Educators *p*-value < 0.05; “Death of patient”—Nurses and Psychiatrists/Training psychiatrists vs PRTs/Educators *p*-value < 0.05; “Workload”—Nurses vs. Psychiatrists/Training psychiatrists and PRTs/Educators and Psychologists/Psychotherapists *p*-values < 0.05; “Team relationship”—post hoc not significant.

**Table 3 brainsci-12-00408-t003:** Descriptive statistics on IES-R and its subscales stratified by type of professional role.

Variable	Total (N = 271)	Nurses (N = 86)	Psychiatrists/Training Psychiatrists (N = 60)	PRTs/Educators (N = 59)	Psychologists/ Psychotherapists (N = 44)	Other Mental Health Professionals (N = 22)	*p*-Value
**IES-R total**							
Mean (SD)	19.38 (15.47)	26.63 (16.86)	15.83 (14.63)	18.73 (13.22)	14.14 (14.11)	12.95 (9.06)	**<0.001**^b^*
Median (Q1, Q3)	17.00 (7.50, 29.00)	24.00 (14.00, 38.00)	13.50 (4.00, 23.25)	16.00 (8.50, 29.50)	11.00 (4.00, 20.25)	13.50 (4.00, 18.00)	
Range	0.00–88.00	0.00–88.00	0.00–77.00	1.00–57.00	0.00–76.00	0.00–29.00	
% IES-R Total score ≥ 33	19.19%	37.21%	10.00%	20.34%	4.55%	0.00%	–
**IES-R avoidance**							
Mean (SD)	6.16 (5.49)	8.43 (6.13)	5.38 (5.29)	5.95 (4.60)	4.11 (4.76)	4.05 (3.93)	**<0.001**^b^*
Median (Q1, Q3)	5.00 (2.00, 9.00)	7.00 (4.00, 11.00)	4.00 (1.00, 8.25)	5.00 (3.00, 8.00)	3.00 (1.00, 5.25)	3.00 (1.00, 6.25)	
Range	0.00–32.00	0.00–32.00	0.00–22.00	0.00–17.00	0.00–24.00	0.00–15.00	
**IES-R intrusion**							
Mean (SD)	7.69 (6.53)	10.60 (7.28)	6.47 (6.19)	7.41 (5.57)	5.50 (5.63)	4.73 (4.11)	**<0.001**^b^*
Median (Q1, Q3)	7.00 (3.00, 11.00)	10.00 (4.25, 16.00)	5.50 (1.00, 10.00)	6.00 (3.00, 10.00)	4.00 (1.00, 9.00)	4.00 (2.00, 7.00)	
Range	0.00–32.00	0.00–32.00	0.00–31.00	0.00–23.00	0.00–29.00	0.00–13.00	
**IES-R hyperarousal**							
Mean (SD)	5.54 (4.94)	7.59 (5.39)	3.98 (4.61)	5.37 (4.59)	4.52 (4.45)	4.18 (3.10)	**<0.001**^b^*
Median (Q1, Q3)	4.00 (2.00, 8.00)	6.00 (4.00, 10.75)	3.00 (0.00, 6.00)	4.00 (2.00, 7.50)	3.50 (1.00, 6.00)	4.00 (2.00, 6.75)	
Range	0.00–24.00	0.00–24.00	0.00–24.00	0.00–17.00	0.00–23.00	0.00–10.00	

^b^ Kruskal test. * Corrected (Hommel method) pairwise comparison result: “IES-R total”—Nurses vs. Psychiatrists/Training psychiatrists, PRTs/Educators, Psychologists/Psychotherapists, and Other mental health professionals *p*-values < 0.05; “IES-R avoidance”—Nurses vs. Psychiatrist/Training psychiatrists, Psychologists/Psychotherapists, and Other mental health professionals *p*-values < 0.01; “IES-R intrusion”—Nurses vs. Psychiatrists/Training psychiatrists, Psychologists/Psychotherapists, and Other mental health professionals *p*-values < 0.01; “IES-R hyperarousal”—Nurses vs. Psychiatrists/Training psychiatrists and Psychologists/Psychotherapists *p*-values < 0.01.

**Table 4 brainsci-12-00408-t004:** Descriptive statistics on DASS-21 subscales stratified by type of professional role.

Variable	Total (N = 271)	Nurses (N = 86)	Psychiatrists/Training Psychiatrists (N = 60)	PRTs/Educators (N = 59)	Psychologists/ Psychotherapists (N = 44)	Other Mental Health Professionals (N = 22)	*p*-Value
**DASS Depression**							
Mean (SD)	5.85 (6.99)	7.60 (7.08)	2.55 (3.10)	6.61 (7.85)	5.13 (7.06)	4.05 (5.87)	**<0.001**^b^*
Median (Q1, Q3)	4.00 (0.00, 8.00)	6.00 (2.00, 10.00)	1.00 (0.00, 4.00)	4.00 (0.00, 10.00)	3.00 (0.00, 6.00)	2.00 (0.00, 6.00)	
Range	0.00–38.00	0.00–28.00	0.00–10.00	0.00–36.00	0.00–38.00	0.00–34.00	
**DASS Anxiety**							
Mean (SD)	3.81 (5.47)	5.77 (6.07)	1.45 (2.48)	4.14 (5.10)	2.30 (5.22)	2.77 (4.94)	**<0.001**^b^*
Median (Q1, Q3)	2.00 (0.00, 6.00)	4.00 (0.50, 8.00)	0.00 (0.00, 2.00)	2.00 (0.00, 6.00)	0.00 (0.00, 2.50)	0.00 (0.00, 4.00)	
Range	0.00–34.00	0.00–26.00	0.00–8.00	0.00–22.00	0.00–34.00	0.00–26.00	
**DASS Stress**							
Mean (SD)	10.84 (7.88)	13.09 (7.82)	7.09 (7.03)	10.54 (7.30)	10.03 (8.43)	9.82 (7.49)	**0.006**^b^*
Median (Q1, Q3)	10.00 (6.00, 16.00)	12.00 (6.00, 19.50)	5.00 (0.50, 10.00)	10.00 (6.00, 15.00)	8.00 (4.00, 14.50)	8.00 (5.50, 14.50)	
Range	0.00–40.00	0.00–30.00	0.00–24.00	0.00–34.00	0.00–40.00	0.00–30.00	

^b^ Kruskal test. * Corrected (Hommel method) pairwise comparison result: “DASS Depression”—Nurses vs. Psychiatrists/Training psychiatrists, Psychologists/Psychotherapists, and Other mental health professionals *p*-values < 0.05; “DASS Anxiety”—Nurses vs. Psychiatrists/Training psychiatrists, Psychologists/Psychotherapists, and Other mental health professionals *p*-values < 0.01, PRTs/Educators vs. Psychiatrists/Training psychiatrists *p*-values < 0.05; “DASS Stress”—Nurses vs. Other mental health professionals *p*-values < 0.01.

**Table 5 brainsci-12-00408-t005:** Variables with a Relative Variable Importance (relVIM) of >20 resulting from the RF for each assessed outcome.

IES-R Total	DASS Depression	DASS Anxiety	DASS Stress
Age	Age	Age	Age
Professional role	Professional role	Professional role	Professional role
Workload	Workload	Workload	Team relationship
Team relationship	Team relationship	Team relationship	Workload
Family divided	Family divided	Family divided	Family divided
LEC13—Severe human suffering (Yes)	LEC9—Other unwanted sexual experience (Yes)	LEC9—Other unwanted sexual experience (Yes)	LEC13—Severe human suffering (Yes)
	LEC13—Severe human suffering (Yes)		LEC9—Other unwanted sexual experience (Yes)
			LEC7—Any stressful event (Yes)

**Table 6 brainsci-12-00408-t006:** Significant predictive factors for moderate/severe symptoms of the outcomes measured (IES-R total, DASS Depression, DASS Anxiety, DASS Stress) identified by multivariable logistic regression analysis.

Variable	OR (95% CI)	*p*-Value
**Moderate/severe symptoms, IES-R total**		
Age	1.06 (1.02–1.10)	0.0037
Professional role (Other health professions)	0.14 (0.02–0.73)	0.0404
Professional role (Psychiatrist/Speciality psychiatry)	0.16 (0.06–0.44)	0.0005
Professional role (Psychologist/Psychotherapist)	0.17 (0.04–0.55)	0.0050
During COVID-19 workload (Increased)	3.41 (1.33–9.30)	0.0128
During COVID-19 team relationship (Improved)	0.22 (0.08–0.59)	0.0034
LEC9—Other unwanted sexual experience (Yes)	7.27 (2.05–28.46)	0.0028
LEC13—Severe human suffering (Yes)	3.34 (1.57–7.43)	0.0022
**Moderate/severe symptoms, DASS Depression**		
Professional role (Psychologist/Psychotherapist)	0.05 (0–0.47)	0.0254
LEC5—Exposure toxic substance (Yes)	60.24 (5.46–990.06)	0.0016
LEC9—Other unwanted sexual experience (YES)	6.62 (1.22–37.32)	0.0273
LEC13—Severe human suffering (Yes)	5.74 (1.69–23.17)	0.0079
**Moderate/severe symptoms, DASS Anxiety**		
Professional role (Psychiatrist/Speciality psychiatry)	0.16 (0.03–0.66)	0.0211
Professional role (Psychologist/Psychotherapist)	0.14 (0.02–0.81)	0.0463
LEC9—Other unwanted sexual experience (Yes)	8.27 (1.67–40.77)	0.0082
**Moderate/severe symptoms, DASS Stress**		
Professional role (Psychiatrist/Speciality psychiatry)	0.30 (0.09–0.89)	0.0388
During COVID-19 team relationship (Improved)	0.23 (0.06–0.72)	0.0168
LEC9—Other unwanted sexual experience (Yes)	6.41 (1.76–24.44)	0.0051
LEC13—Severe human suffering (Yes)	3.03 (1.27–7.53)	0.0142

Cutoff threshold for moderate/severe symptoms: IES-R total ≥ 26; DASS—depression subscale ≥ 14; DASS—anxiety subscale ≥ 10; DASS—stress subscale ≥ 19.

## Data Availability

The data supporting the results and analyses presented in the paper are available on request.

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
