# Peer review of "The Elephant in the Room: A Cross-Sectional Study on the Stressful Psychological Effects of the COVID-19 Pandemic in Mental Healthcare Workers"

_brainsci, 2022, doi:10.3390/brainsci12030408_

Round 1

Reviewer 1 Report

Dear authors,

Thank you for giving me the opportunity to review your original work. I think the manuscript is well written, consists of important new insights in the mental health of MHW and gives breadth and depth to the importance of traumatic life events in relation to COVID19. 

A couple of consideration I would like to make that may help to improve the manuscript. I hope the feedback will make sense. It may sound a bit blunt, but please remember that the work is valuable, original and important support for MHW worldwide!

Title:

Maybe include the design of the title in the analysis?

1.Methods section

In the methods section i could not find the ethical considerations. It would be nice to add your potential ethical review and add the ethics committee decision as a supplementary file

Furthermore, I was wondering whether the authors have preregistered the study and the data management plan

Finally, there is hardly any info on how the authors included the participants. Is there an invitational email in the supplementary material? How did they frame the study to engage potential participants? More details about the selection process is warranted.

2.Results section

It would be helpful to present the cutoff values in the results section with teh tables.

It looks like there are really low scores on mental health dimensions. 

The relation of the association was not clearly expressed in table 5. Maybe this can be added

3.Discussion section

The results are presented in good order and a lot of comparison is made with the literature. However, I do feel that they jump a bit too fast to conclusions. Shouldn't  also be highlighted the other side of the coin; the low levels and lack of proper diagnosis. If we talk about mental health symptoms, there is no diagnosis yet. Furthermore, the cut off values are arbitrary, to say the least and to be fair, at least if I interpret the findings correctly, the level of depression and anxiety is quite low in this study. This also shows us that MHW are resilient and do not have too much mental health burden. Please correct me if I am wrong, but I find it a bit overinterpretation of nice data.

Furthermore, a lot of limitations are not addressed and they should be addressed in order to properly judge the quality of the work

  1. The selection bias can be enormous, depending how the participants were selected and we don;t know the denominator. What is the influence on the interpretation?
  2. Most nurses worked in outpatient settings. That is normally not the setting where you have constant contacts with patients
  3. It is a cross sectional design
  4. all self reported questionnaires
  5. Cut-off values are arbitrary, to say the least.

Conclusion:

Maybe more emphasis on the resilience of the MHWs? I was impressed by the low levels of depression and anxiety. This may also be because they have a very fulfilling profession?

Author Response

ANSWERS TO REVIEWER COMMENTS:

REVIEWER 1:

Dear authors,

Thank you for giving me the opportunity to review your original work. I think the manuscript is well written, consists of important new insights in the mental health of MHW and gives breadth and depth to the importance of traumatic life events in relation to COVID19.

A couple of consideration I would like to make that may help to improve the manuscript. I hope the feedback will make sense. It may sound a bit blunt, but please remember that the work is valuable, original and important support for MHW worldwide!

Title:

Maybe include the design of the title in the analysis? We added the study design in the title.

1.Methods section

In the methods section i could not find the ethical considerations. It would be nice to add your potential ethical review and add the ethics committee decision as a supplementary file. Furthermore, I was wondering whether the authors have preregistered the study and the data management plan. We thank the reviewer for this consideration, however this study is not a clinical trial, so no registration of the study was done, and it was approved by the Ethics Committee of the ASST Spedali Civili of Brescia, Italy (protocol NP4172) as a cross sectional survey.

Finally, there is hardly any info on how the authors included the participants. Is there an invitational email in the supplementary material? How did they frame the study to engage potential participants? More details about the selection process is warranted. An invitation e-mail was initially sent to potential participants and further participants were recruited using a snowball sampling procedure, by asking receivers to disseminate the survey link received by e-mail or by WhatsApp link among their colleagues. More detailed information was added in the Methods section. Also, a file was added as Supplementary material (Supplementary file – Invitation text), in its original version in Italian language and translated into English language. 

2.Results section

It would be helpful to present the cutoff values in the results section with the tables. It looks like there are really low scores on mental health dimensions. The relation of the association was not clearly expressed in table 5. Maybe this can be added. We thank the reviewer for the suggestion. We added a note as requested.

3.Discussion section

The results are presented in good order and a lot of comparison is made with the literature. However, I do feel that they jump a bit too fast to conclusions. Shouldn’t also be highlighted the other side of the coin; the low levels and lack of proper diagnosis. If we talk about mental health symptoms, there is no diagnosis yet. Furthermore, the cut off values are arbitrary, to say the least and to be fair, at least if I interpret the findings correctly, the level of depression and anxiety is quite low in this study. This also shows us that MHW are resilient and do not have too much mental health burden. Please correct me if I am wrong, but I find it a bit overinterpretation of nice data.

Furthermore, a lot of limitations are not addressed and they should be addressed in order to properly judge the quality of the work. 1 The selection bias can be enormous, depending how the participants were selected and we don’t know the denominator. What is the influence on the interpretation? 2 Most nurses worked in outpatient settings. That is normally not the setting where you have constant contacts with patients 3 It is a cross sectional design 4 All self reported questionnaires 5 Cut-off values are arbitrary, to say the least. We thank the reviewer for these important considerations. We have reevaluated the limitations and strengths of our study and we have added a more extensive and complete explanation concerning these points in the Discussion section.  

Conclusion: Maybe more emphasis on the resilience of the MHWs? I was impressed by the low levels of depression and anxiety. This may also be because they have a very fulfilling profession? We have added an observation about this issue in the Discussion section and, accordingly with the changes made in the discussion section, we changed also the conclusion section.

Reviewer 2 Report

Please follow CROSS: https://link.springer.com/content/pdf/10.1007/s11606-021-06737-1.pdf

18 participant characteristic missing..

29 might represent

51 I also suggest adding a reference to the employment relationship and context: "Moreover, the importance of a healthy psychosocial work environment to enhance job satisfaction of all health professionals and to avoid role conflict and burnout syndrome during the COVID-19 pandemic should not be underestimated."
(ref: https://pubmed.ncbi.nlm.nih.gov/34574600/ )

110 How was it prepared and provided?

143 mean±SD and median[IQR]

156 “A Random Forest method was used for a regressional machine learning approach” ..

157 Random forest is not parametric? Why?

164 Unfortunately this choice creates an intrinsic bias .. because the age values are innumerable while the occupancy values and the others are smaller (1-3) .. I already expect to know that the age value is the most influential .. Unfortunately if you want to keep the construct in the manuscript you have to try to make all values become similar (dichotomous .. for example age becomes less or greater than the median).

If you do not choose this option, remove this approach from results and discussion

Too voluminous conclusions with too restrictive limitations ..

I recommend 10-15 lines maximum of conclusion. I recommend removing phrases like "Our study represents an important contribution to the recent literature". Obviously I hope so for the authors, but better not be pretentious.

Author Response

ANSWERS TO REVIEWER COMMENTS:

REVIEWER 2:

Please follow CROSS: https://link.springer.com/content/pdf/10.1007/s11606-021-06737-1.pdf We thank the reviewer for this comprehensive and very useful proposal of reporting of survey studies, which includes detailed standard reporting checklist for survey research. We have taken into consideration this reporting system.           

18 participant characteristics missing. We added some participant characteristics in the abstract as requested.

29 might represent We have added the word “might”, as suggested, in the Abstract.

51 I also suggest adding a reference to the employment relationship and context: "Moreover, the importance of a healthy psychosocial work environment to enhance job satisfaction of all health professionals and to avoid role conflict and burnout syndrome during the COVID-19 pandemic should not be underestimated." (ref: https://pubmed.ncbi.nlm.nih.gov/34574600/) We have added the reference, as suggested, in the Introduction section.

110 How was it prepared and provided? More detailed information about the methods, particularly the recruitment process, were included in Methods section.

143 mean±SD and median [IQR] We corrected the text.

156 “A Random Forest method was used for a regressional machine learning approach”. We changed the text accordingly.

157 Random Forest is not parametric? Why? Random Forest is a non-parametric method in which the data do not come from prescribed models determined by a small number of parameters.

164 Unfortunately this choice creates an intrinsic bias because the age values are innumerable while the occupancy values and the others are smaller (1-3). I already expect to know that the age value is the most influential. Unfortunately, if you want to keep the construct in the manuscript you have to try to make all values become similar (dichotomous for example age becomes less or greater than the median). If you do not choose this option, remove this approach from results and discussion Random Forest (RF) is a well-suited method able to deal with variables of different nature, quantitative and qualitative with few categories. Consequently, it provides good results without transforming age in a dichotomous variable (see statistical analysis section). Moreover, the Variable Importance Measure used in this study for identifying most important predictors of the outcomes (IES-R total, DASS scale and subscale) does not provide biased results with variables of different nature. We changed the text accordingly adding also the reference number (27).

Too voluminous conclusions with too restrictive limitations. I recommend 10-15 lines maximum of conclusion. I recommend removing phrases like "Our study represents an important contribution to the recent literature". Obviously, I hope so for the authors, but better not be pretentious. We thank the reviewer for this comment. We have reevaluated and restructured the limitations and strengths of our study, and we have restructured the conclusion section (see Discussion and Conclusion sections).

Reviewer 3 Report

Brief summary

The paper entitled “The elephant in the room: the stressful psychological effects of 2 COVID-19 pandemics in mental healthcare workers” by Minelli et al., aims to evaluate the possible psychological effects of COVID-19 diffusion on healthcare workers. To this aim, an online survey was administered to participants.

Generally, I believe that the authors will agree that it is useful to believe that mental health professionals do not present risks of psychopathological vulnerability in order to be effective in their profession. The findings of this study could be useful in demonstrating the need to guarantee psychological assistance services also for mental health workers. In addition, the work could help the governors to understand the importance to conduct research above all in clinical context, so also outside the laboratories.

Although the paper has been carefully prepared, I have some questions for the authors that require major attention:

- The authors declare that they want to investigate the possible influence of previous stressful events on the emotional distress caused by the pandemic. Healthcare workers who have experienced stressful events in the past were more exposed to the emotional consequences of the pandemic. I agree with this hypothesis, but I wonder if the symptoms of anxiety and depression were not already present in the participants before the pandemic event.

- Some data for me are not clear: being a psychologist or a psychiatrist seemed to be a protective factor. How many health professionals who reported unwanted sexual intercourse were psychologists and psychiatrists? Knowing that 10% of people have experienced this type of stressful event is very disturbing. In the future, the authors may consider doing a study solely on this aspect.

- The LEC-5 test they administered refers to the entire lifespan. Did they consider that people who have suffered trauma may have been more affected than those who suffered it several years ago? Have they considered the treatment variable? It is possible that people who have faced highly stressful life events have undergone treatments following these events. Is the type of profession carried out to be a protective factor or the possibility of having done a course of therapy in the past? Was this data collected?

PARTICIPANTS:

  • The authors define Piedmont, Lombardy, and Veneto as regions with the greatest spread of the COVID-19 virus, but do not report any data relating to the percentage of contagion in these territories. They also do not clarify which period they are referring to. When was there the greatest impact? During the first or second wave? This statement has to be supported with proper data reporting the mortality of covid 19, normalized with the population, at varying the Italian regions. The authors could represent the required data with a graph for the first wave and a graph for the second wave.
  • Why do the authors use as a criterion of exclusion the inability to read and write in Italian? Are there illiterate health care workers?

Other comments

  • Line 39: please, add the specific object of the two waves: COVID-19.
  • Line 88: what do the authors mean with COVID-19 variables? Specific it, please.
  • Why do authors entitle the paper "the elephant in the room"?

Author Response

ANSWERS TO REVIEWER COMMENTS:

REVIEWER 3:

The paper entitled “The elephant in the room: the stressful psychological effects of 2 COVID-19 pandemics in mental healthcare workers” by Minelli et al., aims to evaluate the possible psychological effects of COVID-19 diffusion on healthcare workers. To this aim, an online survey was administered to participants.

Generally, I believe that the authors will agree that it is useful to believe that mental health professionals do not present risks of psychopathological vulnerability in order to be effective in their profession. The findings of this study could be useful in demonstrating the need to guarantee psychological assistance services also for mental health workers. In addition, the work could help the governors to understand the importance to conduct research above all in clinical context, so also outside the laboratories.

Although the paper has been carefully prepared, I have some questions for the authors that require major attention:

- The authors declare that they want to investigate the possible influence of previous stressful events on the emotional distress caused by the pandemic. Healthcare workers who have experienced stressful events in the past were more exposed to the emotional consequences of the pandemic. I agree with this hypothesis, but I wonder if the symptoms of anxiety and depression were not already present in the participants before the pandemic event. We thank the reviewer for this comment. We have modified the limitations and strengths of our study and we have added a more extensive and complete section concerning these points in the Discussion, including the possible presence of anxiety and depressive symptoms prior to the pandemic occurrence, possibly limiting the interpretation of the results (see Discussion and Conclusion sections).

- Some data for me are not clear: being a psychologist or a psychiatrist seemed to be a protective factor. How many health professionals who reported unwanted sexual intercourse were psychologists and psychiatrists? Knowing that 10% of people have experienced this type of stressful event is very disturbing. In the future, the authors may consider doing a study solely on this aspect. As suggested by the reviewer we changed the Supplementary Table 1.

- The LEC-5 test they administered refers to the entire lifespan. Did they consider that people who have suffered trauma may have been more affected than those who suffered it several years ago? Have they considered the treatment variable? It is possible that people who have faced highly stressful life events have undergone treatments following these events. Is the type of profession carried out to be a protective factor or the possibility of having done a course of therapy in the past? Was this data collected? Thanks for this comment. The LEC-5 is aimed to assess the exposure to a range of possible traumatic/stressful events throughout a person's life. In our study we did not add questions regarding the period of exposure to the traumatic/stressful event and whether the person received any treatment following these events, as such information was beyond the scope of our study. We agree with the Reviewer that is important to deepen the possible association of such variables with psychological distress in MHWs. We added this point as an implication for future studies in our manuscript, in the Discussion section.

PARTICIPANTS:

The authors define Piedmont, Lombardy, and Veneto as regions with the greatest spread of the COVID-19 virus, but do not report any data relating to the percentage of contagion in these territories. They also do not clarify which period they are referring to. When was there the greatest impact? During the first or second wave? This statement has to be supported with proper data reporting the mortality of covid 19, normalized with the population, at varying the Italian regions. The authors could represent the required data with a graph for the first wave and a graph for the second wave.

Why do the authors use as a criterion of exclusion the inability to read and write in Italian? Are there illiterate health care workers? We thank the reviewer for the comments. This is a mistake. The study was performed exclusively in Italian language and the inability to read and write in Italian was indeed applied as an exclusion criterion. To avoid misinterpretation, we removed this phrase from the methods section.

Other comments

Line 39: please, add the specific object of the two waves: COVID-19. We have inserted a figure in the supplementary files and specified in the introduction when the two first waves occurred in the North of Italy (from the end of February to the end of March 2020 and from the end of October to the end of November 2020, respectively).

Line 88: what do the authors mean with COVID-19 variables? Specific it, please. The COVID-19 variables are the ones specified in Table 2: being COVID-19 positive, family member positive for COVID-19, fear of contagion, death a loved one by COVID-19 infection, death a patient by COVID-19 infection, changes in workload, team relationship, family divided due to the current pandemic. We have explained the variables in the Introduction section.

Why do authors entitle the paper "the elephant in the room"? The title refers to the usual expression “elephant in the room” for a metaphorical example of a serious problem or a difficult situation that people are usually aware of, but often ignore or choose not to mention, face or discuss. We wanted to highlight the importance of considering all the psychological consequences imposed by the pandemic and risk factors for emotional distress to understand their real impacts on mental healthcare workers. We have added some lines explaining the choice of this expression and the connection of the title with our findings in the end of the Conclusion section.

Round 2

Reviewer 2 Report

I can suggest acceptance in the present form

Reviewer 3 Report

Congratulations to the authors who have reviewed their work following the requested suggestions. In this difficult historical period, it is important to carry out research work on health professionals who are at the center of local and national attention. Studies like this could help local governors design interventions that promote the well-being of health care workers.